# Micro-Three-Coil Sensor with Dual Excitation Signals Use Asymmetric Magnetic Fields to Distinguish between Non-Ferrous Metals

**DOI:** 10.3390/s23031637

**Published:** 2023-02-02

**Authors:** Jiaju Hong, Yucai Xie, Shuyao Zhang, Haotian Shi, Yu Liu, Hongpeng Zhang, Yuqing Sun

**Affiliations:** 1Marine Engineering College, Dalian Maritime University, Dalian 116026, China; 2Navigation College, Dalian Maritime University, Dalian 116026, China

**Keywords:** micro-three-coil, asymmetric magnetic, oil detection, non-ferrous metals

## Abstract

Intelligent operation and maintenance technology for vessels can ensure the safety of the entire system, especially for the development of intelligent and unmanned marine technology. The material properties of metal abrasive particles in oil could demonstrate the wear areas of the marine mechanical system because different components consist of different materials. However, most sensors can only roughly separate metallic contaminants into ferromagnetic and non-ferromagnetic particles but cannot differentiate them in greater detail. A micro-three-coil sensor is designed in this paper; the device applies different excitation signals to two excitation coils to differentiate materials, based on the different effects of different material particles in the asymmetric magnetic field. Therefore, a particle’s material can be judged by the shape of the induction electromotive force output signal from the induction coil, while the particle size can be judged by the amplitude of the signal. Experimental results show that the material differentiation of four different types of particles can be achieved, namely, of aluminum, iron, 304 stainless steel, and carbon steel. This newly designed sensor provides a new research prospect for the realization of an inductive detection method to distinguish non-ferrous metals and a reference for the subsequent detection of metal contaminants in oil and other liquids.

## 1. Introduction

Intelligent operation and maintenance techniques are used to provide feedback on the state of a vessel by testing the vessel’s mechanical systems. The wear of the system can be determined by testing the cleanliness of the oil [1]. Data show that around 75% of hydraulic system failures, 35% of diesel engine operation failures, 38.5% of gear failures, and 40% of rolling bearing failures are caused by abrasive particles in the oil [2]. Therefore, the detection of the size, type, and quantity of abrasive particles in the oil is very important for the safety of vessels. The operation of a vessel’s machinery system includes the break-in phase, normal operation phase, and wear phase. When the system enters the wear phase, damaged equipment and lubricant should be replaced in a timely manner. Therefore, judging the degree of wear and determining the damaged parts is the focus of equipment condition monitoring [3,4,5].

However, vessels generally sail offshore for months before they can be brought back to dock for laboratory tests of oil. At present, for oil cleanliness testing studies, a laboratory delivery method is often used, collecting lubricants from operating equipment and sending them to a laboratory for oil quality analysis through methods such as iron spectrum analysis [6,7], ultrasonic testing, and optical testing [8,9]. Results of the testing arrive after a delay and the method cannot monitor equipment status in real-time. Therefore, an inductive detection method, based on a detection system for metal abrasive particles in the oil, has gained wide attention. This method has a high tolerance for oil translucency and can distinguish between metal abrasive and non-metal abrasive particles. The inductive detection sensors most commonly used nowadays include single coil sensors, dual coil sensors, and traditional three coil sensors. Zhu et al. [10] and Shi et al. [11] improved the detection accuracy of sensors by adding high permeability material inside the single coil sensor and concentrating the magnetic field of the coil. L DU et al. [12] and Yu et al. [13] improved the detection accuracy of a single coil sensor with an LC resonant circuit. Shi et al. [14] made a magnetic suction oil detection sensor by adding a magnet to a single coil, which can achieve both inductive and capacitive detection. Xie et al. [15] used a dual coil sensor, constructed it as an AC bridge, and designed a circuit to extract the voltage signal. Zeng et al. [16] used two coil building blocks in a dual mode oil detection sensor with inductive detection and capacitive detection. Ren et al. [17,18] and Qian et al. [19] designed self-balancing circuits to balance the voltage difference between the two sides of a three-coil sensor and used shielding materials to further improve the detection accuracy and flux of the sensor. Additionally, Qian et al. [20] designed a special shielded coil which can suppress the interference of dielectric components and improve the detection accuracy of a sensor. Ran et al. [21] combined an LC resonant circuit with a three-coil sensor to improve the flux variation of the induction coil and expand the induction electromotive force of the sensor.

There are many different device materials used in different mechanical systems, and a more detailed distinction between the materials of metal abrasive particles would allow for quicker location of faulty devices, reducing repair time and costs. In the abovementioned sensors, the output signal of single-coil sensors and dual-coil sensors are generally single pulse, using the direction of the pulse signal to distinguish between ferromagnetic particles and non-ferromagnetic particles. Ferromagnetic particles contain many different materials. Thus, when iron particles and carbon steel particles pass through the sensor simultaneously, it is impossible to distinguish between the two types of particles simply through the output signal of these two sensors, as their particles have the same properties. The material distinction of the properties of these two particles can only be identified by changing the frequency or algorithm. The single output signal of the traditional three-coil sensor has only one peak and one trough and can only identify metal abrasive particles as either ferromagnetic particles or non-ferromagnetic particles, according to their phase. They cannot perform a more detailed distinction, a fact which is not conducive for locating damaged parts at a later stage. In this study, we apply microfluidic technology to design a micro-triple-coil sensor and conduct different excitation signals in two excitation coils to constitute an unbalanced magnetic field so that particles of different properties act differently in the magnetic field and achieve the differentiation of metal abrasive particles with different materials of the same property through multi-magnetic field coupling.

## 2. Sensor Model

Figure 1 illustrates the schematic diagram of a miniature three-coil sensor. The coils on both sides of the exterior are identical excitation coils and are wound in opposite directions so that when coil 1 and coil 3 are excited by AC excitation sources of different frequencies, the excitation signal strengths of the two excitation coils are different and the magnetic fields generated by the two excitation coils are coupled to each other, producing asymmetric alternating magnetic fields in opposite directions. The direction of the alternating magnetic field generated by the two excitation coils is closely related to the phase of the excitation signal. The magnetic fields at the positive center of the detection coils cannot cancel each other out; particles of different materials produce induced potentials of different shapes and amplitudes in the coils.

The schematic diagram of the micro-three-coil sensor is shown in Figure 2. Metal particles in an alternating magnetic field produce magnetization and eddy current effects, causing changes in the inductance and impedance of the coil [22,23,24,25,26].

When two excitation coils are subjected to different frequencies of excitation signals,
(1)$$U_{out}=\frac{\omega \left(M_{13}U_{in1}-M_{23}U_{in2}\right)}{\sqrt{{\left(R_{1}+R_{2}\right)}^{2}+\omega^{2}{\left(L_{1}+L_{2}-2M_{12}\right)}^{2}}}$$
where *U*_1_ and *U*_2_ represent the AC signals applied to the two excitation coils; *R*_1_, *R*_2_, and *R*_3_ are the intrinsic resistances of the coils; and *M*_12_, *M*_13_ and *M*_23_ are the mutual inductances between the three coils.

Figure 3 illustrates the magnetic field density mode distribution of the coil under the excitation of a sinusoidal signal with a frequency of 1.15 MHZ and voltage of 5 V, and frequency of 2 MHZ and voltage of 5 V using the COMSOL software. The number of turns of the coil is 200, the diameter of the inner hole is 2.5 mm, and the diameter of the outer hole is 5 mm.

As shown in Figure 3, when the excitation frequency becomes too high, the AC resistance of the coil increases, and the magnetic field’s strength decreases instead. At different excitation frequencies, the magnetic field strength of the coil is different, and the particles with different properties are subjected to different magnetization and eddy current effects in the magnetic field due to their different relative permeability and electrical conductivity. This leads to different response curves of the coil to the particles, and the differentiation of nonferrous metals can be achieved according to the response characteristics of the particles in the asymmetric magnetic field.

## 3. Results and Discussions

### 3.1. Subsection

Figure 4 introduces the overall experimental setup and sensor. The two excitation coils and induction coils used in the three-coil sensor are identical and wound from copper wire with a wire diameter of 70 μm. The excitation and induction coils are made of the same size coils; excitation coil 1 and excitation coil 2 are wound in reverse. The number of turns of the coils is 200, with a 2.5 mm inner hole diameter and a 5 mm outer hole diameter. Three coils are placed side by side; the middle coil is the induction coil and the two side coils are the excitation coils. A copper strip of 2.5 mm diameter is used to pass through the coils and is fixed on the slide and placed into the mold. Polydimethylsiloxane (PDMS) solution with a 10:1 curing agent was poured into the mold and placed in a vacuum drying oven for 30 min; after the PDMS was cured, the copper strip was removed and the micro-three-coil sensor was made. A linear DC power supply is used to supply power to the signal processing circuit, and the excitation signals of different frequencies and amplitudes are provided to the two excitation coils by the waveform generator. During the experiment, the particles to be measured are fixed on the microfiber rope, and the particles are controlled to pass through the micro-three-coil sensor at a uniform speed by the slide table.

Since AC excitation is used, the output induction signal is also a sinusoidal AC signal, which is not convenient for subsequent acquisition and processing, so a signal conditioning circuit needs to be designed to convert it into a DC signal. As shown in Figure 5, this paper processes the output induction potential by designing half-wave rectification, low-pass filtering, Butterworth low-pass filter, and post-amplification circuits. The AC signal is converted into a pulsating DC signal by the half-wave rectification circuit, and then the high-frequency noise is filtered out by the low-pass filtering circuit, at which time the signal amplitude is small; therefore, it is amplified by the amplification circuit. The output signal is acquired through a data acquisition card.

### 3.2. Micro-Three-Coil Sensor Detects Particles

Copper particles are darker in color. Thus, a simple distinction can be made by color, while the color of the oxidized aluminum particles is closer to other ferromagnetic particles, which is not easy to distinguish. Therefore, in this study, aluminum particles are selected as non-ferromagnetic particles for experiments. To make the signal effect obvious and avoid signal errors caused by external noise, larger particles are selected for experiments and analyses, which can reduce the interference of noise on the output curve and facilitate the observation of the essential characteristics of the signal.

In this experiment, the size of the particles to be measured is above 300 μm, because large particles respond strongly to the magnetic field and are not easily disturbed by external noise. Noises can distort the signal curve of the response. Since this paper aims to distinguish the particle properties instead of improving accuracy, smaller particles are not used.

After several experimental analyses, a sinusoidal excitation signal with an amplitude of 5 V at 1.15 MHz was selected as the final excitation signal. The same excitation signal is applied to both excitation coils to investigate the output signals of ferromagnetic and non-ferromagnetic metal particles in a symmetric magnetic field. Particles of about 700 μm were selected for the experiment and the experimental results are shown in Figure 6. When particles of different materials pass through the sensor in the same direction, the phase of output signal of the three-coil sensor is related to the direction of the particle passage and the direction of the excitation signal. Therefore, the signal phase of aluminum metal particles is opposite to that of iron materials such as iron particles, 304 stainless steel particles, and carbon steel particles. Figure 6 shows the output results of the miniature triple coil when the same excitation signal is used and it can be seen that the material differentiation of metals cannot be achieved using only a single excitation.

### 3.3. Detection of Aluminum Particles Using a Dual Excitation Micro-Three-Coil Sensor

In this experiment, aluminum particles of 350 μm, 500 μm, 620 μm, and 710 μm were selected for the experiment; the particles to be tested were fixed on a microfiber rope with glue and the particles were controlled by the track to pass through the sensor. The electrical conductivity of the aluminum particles was 3.774 × 10^7^ S/M and the relative magnetic permeability was 1.000022. The excitation signal of excitation coil 1 was selected as 1.15 MHZ with a sinusoidal excitation signal of 5 V, and the excitation signal of excitation coil 2 was selected as 2 MHZ with a sinusoidal excitation signal of 5 V. The signal plots of the output of different sizes of aluminum particles passing through the asymmetric magnetic field were selected separately and are shown in Figure 7. The aluminum particles are subject to the eddy current effect in the alternating magnetic field, which weakens the magnetic field of the excitation coil. The initial phase generated by the aluminum particles is a positive phase, which produces two positive pulses and one negative pulse. The maximum wave peak value is similar in size to the wave trough value, and the signal, at this time, is different from the output of the micro-triple-coil sensor with the same excitation signal. According to the experimental results, the positive pulse value of 710 μm aluminum particles is 3.174 mV, the negative pulse value is 3.892 mV, and the fundamental noise of the circuit is 0.8 mV.

### 3.4. Detection of Iron Particles Using a Dual Excitation Micro-Three-Coil Sensor

Iron particles with particle sizes of 600 μm, 738 μm, 800 μm, and 952 μm were selected for the test. The relative magnetic permeability of iron particles is 4000 and the electrical conductivity is 1.12 × 10^7^ S/M. Figure 8 illustrates that when iron particles are affected by the external magnetic field, the atomic magnetic moments within the particles are neatly arranged in the same direction and the particles will exhibit certain magnetic properties, which are the magnetization phenomena of the particles. The iron particles under control of the slide rail pass through the sensor and produce a magnetization effect and eddy current effect in the alternating magnetic field. Since the magnetization effect has a greater effect on the magnetic field than the eddy current effect, the iron particles enhance the original magnetic field, causing two negative pulses in the output signal. The size difference of iron particles can be identified through signal amplitude.

### 3.5. Detection of 304 Stainless Steel Particles Using a Dual Excitation Micro-Three-Coil Sensor

304 stainless steel is a non-magnetic material, but it turns magnetic after cold working deformation when part of the austenitic tissue undergoes martensitic phase transformation [27,28]. According to the literature [28], the electrical conductivity of stainless steel is 1.43 × 10^6^ S/M and the relative magnetic permeability is 1.09. Therefore, as shown in Figure 9, when performing 304 stainless steel particle detection, the initial phase produced by the particles is the same as that of iron particles due to the magnetic nature of the processed particles. Due to the difference in relative magnetic permeability and electrical conductivity of the material, 304 stainless steel particles have a different effect on the alternating magnetic field than aluminum or iron particles. Therefore, stainless steel only produces a negative pulse signal. When the particle size is larger than 850 μm, a positive pulse signal will be generated. The main reason for this phenomenon is that the increase in particle size enhances the effect on both excitation coils, which act in opposite directions, so that forward pulses will start to appear.

### 3.6. Detection of Carbon Steel Particles Using a Dual Excitation Micro-Three-Coil Sensor

Carbon steel is a ferromagnetic material and its tribological properties are significantly influenced by the magnetic field. Therefore, the use of carbon steel as a friction pair material is more favorable for studying the mechanism of the influence of the magnetic field on the tribological properties of ferromagnetic materials [29].

Iron particles with particle sizes of 400 μm, 614 μm, 710 μm, and 1010 μm were selected for the tests. The electrical permeability of carbon steel particles varied between 3.3 × 10^7^ S/M and 5.73 × 10^7^ S/M, and the relative permeability was 150. This study has suggested that the coil’s magnetic field becomes saturated when the relative permeability is larger than 100. After this, an increase in relative permeability had a weakening effect on the magnetic field of the aggregation, so the relative permeability of carbon steel differs from that of iron; however, the difference in the enhancement effect on the magnetic field is small. Figure 10 illustrates the experimental results. When carbon steel particles pass through the sensor, the initial phase of the signal is −180°, the output signal of the induction coil has two negative pulses and one positive pulse, the wave peak value is 4.593 mV for 710 μm carbon steel particles, and the wave trough value is 6.582 mV.

## 4. Discussion

During the experiment, we found that the phase of the excitation signal has a great influence on the shape of the output signal. The main reason for this is that the non-uniform magnetic field distribution changes due to the different phases of the excitation signal. Further research and consideration of the theoretical model between the phase and the signal is required.

## 5. Conclusions

Figure 11 depicts a comparison of the detection signals of a conventional three-coil sensor, micro-three-coil sensor, and micro-three-coil sensor with different excitation signals. Conventional three-coil sensors and micro-three-coil sensors can only distinguish ferromagnetic particles from non-ferromagnetic particles. In our study, however, by applying different excitation signals to the two excitation coils of a micro-three-coil sensor and constructing an asymmetric magnetic field, material differentiation of different types of particles was achieved.

In an asymmetric magnetic field, the initial phase of aluminum particles is a positive phase which produces two positive pulse signals and one negative pulse signal. The initial phase of iron particles, 304 stainless steel particles, and carbon steel particles is a negative phase, whereas iron particles produce two negative pulse signals, 304 stainless steel particles produce one positive pulse signal plus one negative pulse signal, and carbon steel particles produce two negative pulse signals plus one positive pulse signal. Therefore, in this paper, particle type can be distinguished by the pattern of the output signal alone and the particle size can be distinguished by the amplitude of the signal.

Through experiment, metal materials can be effectively distinguished when multiple excitation signals are used. However, this method discards the detection accuracy of the sensor, so further optimization of the coil size, excitation signal, and signal processing circuit of the sensor is needed in subsequent research in order to improve its detection accuracy under the premise of distinguishing particle properties effectively. The technology provides a new reference for signal detection on the basis of intelligent vessel operation and maintenance, enabling the detection of a wide range of pollutants, which is a very important reference for the determination of the location of multi-ship faults.

## Figures and Tables

**Figure 1 sensors-23-01637-f001:**
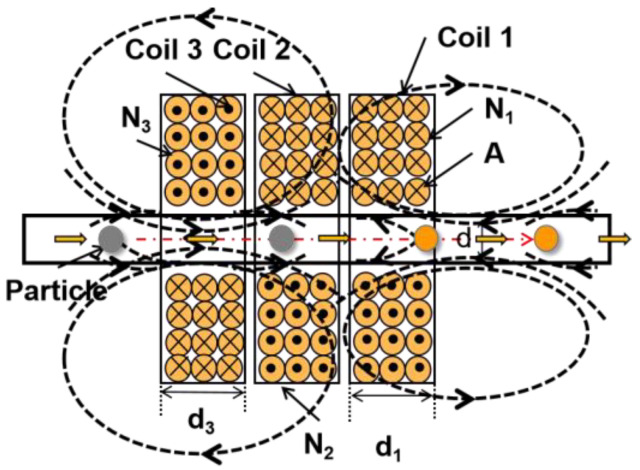
Schematic diagram of the sensor.

**Figure 2 sensors-23-01637-f002:**
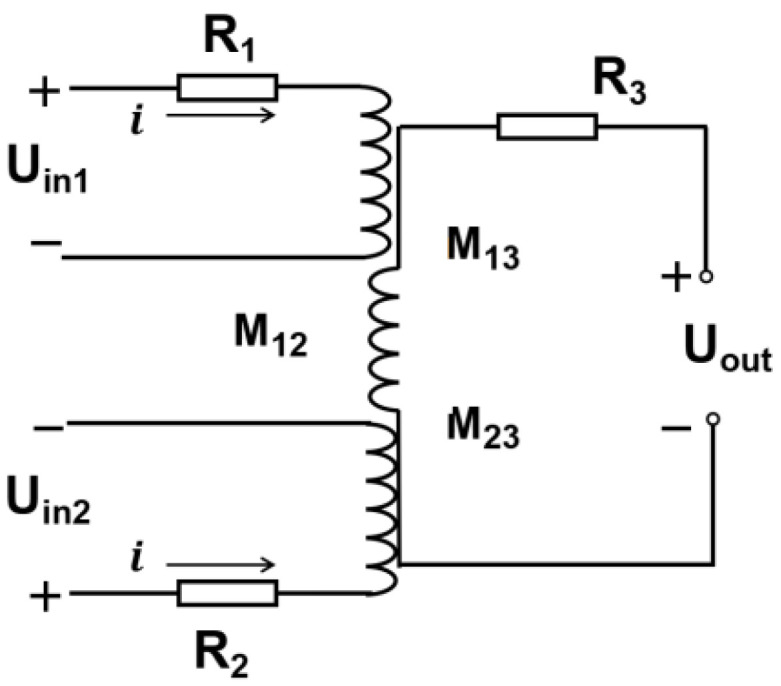
Sensor schematic.

**Figure 3 sensors-23-01637-f003:**
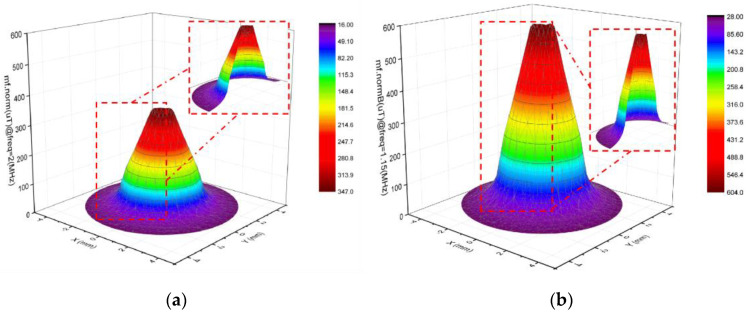
Magnetic field density mode distribution of the excitation coil under different excitations. (**a**) The excitation frequency is 2 MHz; (**b**) the excitation frequency is 1.15 MHz.

**Figure 4 sensors-23-01637-f004:**
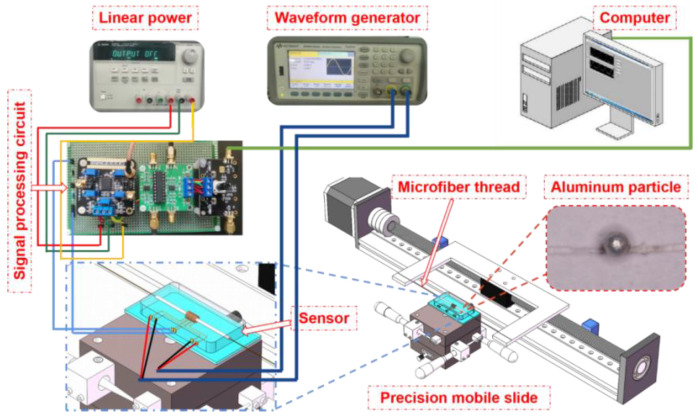
Sensor and experimental system.

**Figure 5 sensors-23-01637-f005:**
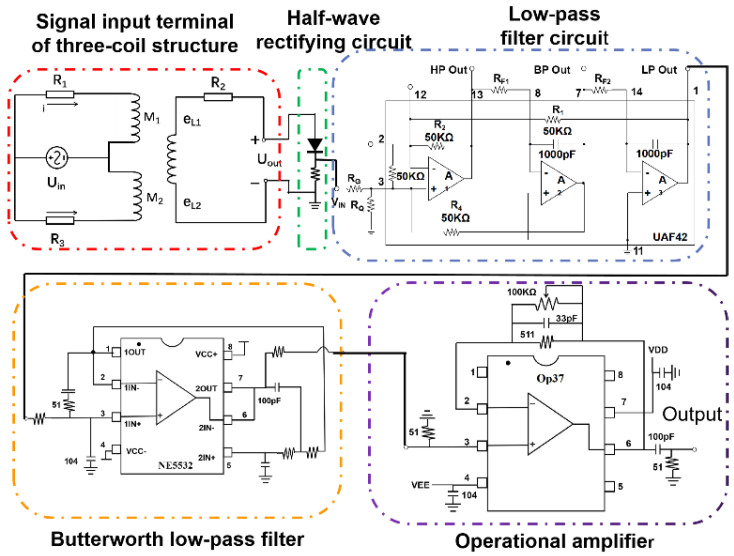
Signal processing circuit diagram.

**Figure 6 sensors-23-01637-f006:**
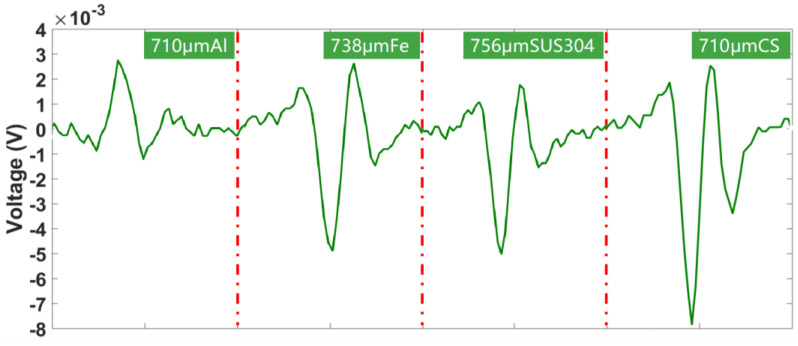
Inductive electric potential generated by particles of different materials.

**Figure 7 sensors-23-01637-f007:**
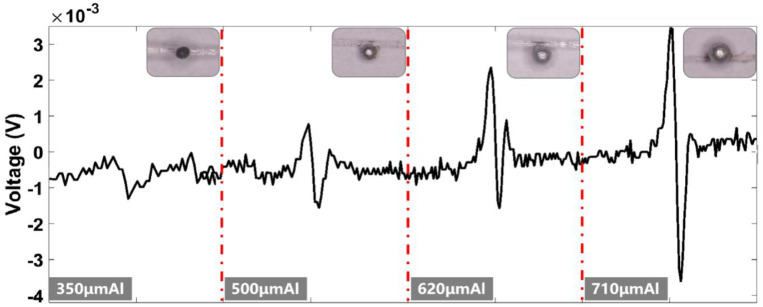
Inductive electric potential generated by different particle sizes of aluminum particles.

**Figure 8 sensors-23-01637-f008:**
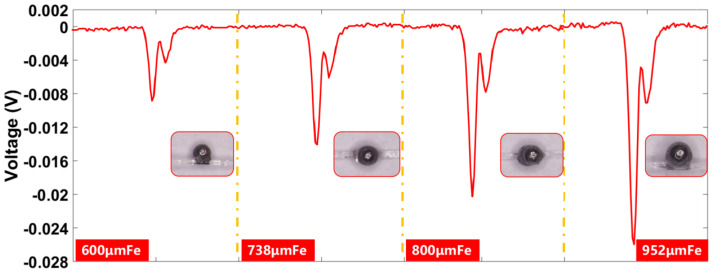
Inductive electric potential generated by iron particles of different particle sizes.

**Figure 9 sensors-23-01637-f009:**
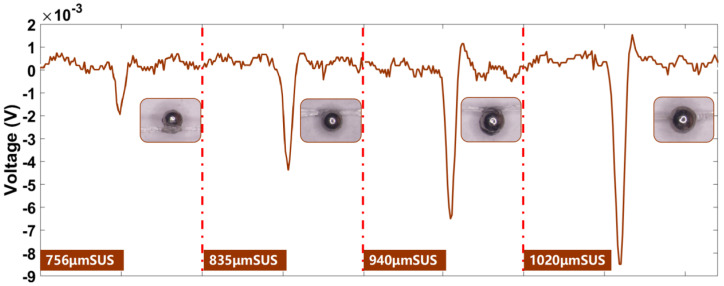
Inductive electric potential generated by 304 stainless steel particles of different particle sizes.

**Figure 10 sensors-23-01637-f010:**
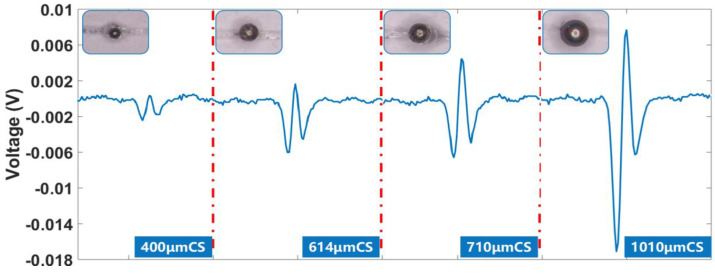
Inductive electric potential generated by carbon steel particles of different particle sizes.

**Figure 11 sensors-23-01637-f011:**
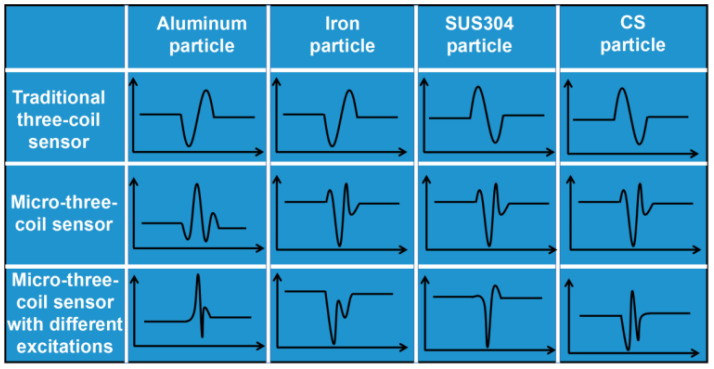
Detection method and signal.

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
