# Peer review of "Micro-Three-Coil Sensor with Dual Excitation Signals Use Asymmetric Magnetic Fields to Distinguish between Non-Ferrous Metals"

_sensors, 2023, doi:10.3390/s23031637_

Round 1

Reviewer 1 Report

The authors designed a three-coil sensor and created an non-uniform magnetic field magnetic field by applying two different excitation signals in two excitation coils to achieve the purpose of non-ferrous metal differentiation. The paper is very innovative and informative and can be acceptted after minor revision, but there are still some questions need to be answered by the authors.

(1) Comparison between micro triple coils and conventional triple coils. Where do the conventional curves come from? Which literature presents these curves?

(2) Why the two peak appear in the output signal comparing with traditional sensors?

(3) What are the advantages of this paper compared with other research results

(4) The authors can verify the particle concentration over time in a subsequent study. Accurate counting of wear debris is of equal importance as size measurement.

(5) Please indicate in the simulation model the parameters of the coil, the amplitude and frequency of the loading signal of the excitation coil, and the speed of the metal particles through the coil.

Reviewer 2 Report

In this paper, a Dual Excitation Micro-three-coil Sensor is designed to distinguish four non-ferrous metal particles. The sensor constitutes an asymmetric magnetic field by conducting different excitation signals in two excitation coils. By contrasting the Micro-three-coil Sensor, this article mainly studied the detection signal pattern of the Dual Excitation Micro-three-coil Sensor on four metal particles, and analyzed the amplitudes of different particle sizes. The results have certain guiding significance. I think this manuscript and the study design need to be improved. The paper is somewhat novel and accepted for publication after revision.

Several concerns are raised below.

1. This paper lacks the theoretical part for the sensor to detect and distinguish four metal particles, and suggests adding some theories.

2. The first paragraph of the introduction is lengthy and it is recommended to be concise.

3. Page 3, the description of Figure 1 is too simple and inaccurate. Whether the circle in Figure 1 represents magnetic field lines, if so, the direction of the magnetic field lines drawn does not match the direction of the coil current, please explain;

4. Page 3, R1 in Figure 2 is inconsistent with r1 in Equation, and the parameters are not explained after the formula.

5. Page 3, Figure 3 is a magnetic flux density distribution of the coil, please indicate what software is used to draw it.

6. In Figure 3-(b) freq=2 in the ordinate, which is inconsistent with the frequency 1.15 stated in the text, please explain.

7. The 2. Sensor Model part of the text suggests following the principle of first article and then the diagram.

8. In 3.1, Why do metal particles pass through the sensor at a constant speed? Please explain the detection principle of this part.

9. In 3.2, I suggest the authors to compare the Micro-three-coil Sensor with the Dual Excitation Micro-three-coil Sensor.

10. Page 5, 3.2, paragraph 3: “a sinusoidal excitation signal with an amplitude of 5 V at 1.15 MHz was selected as the final excitation signal”, what is the experimental or theoretical basis for the choice of excitation source?

11. In 3.3-3.6,how are the frequency parameters of the two excitation coils determined? What is the basis for the diameter of the metal particles?

12. Page 7, 3.5: “Therefore, stainless steel only produces a negative pulse signal, and when particle size is larger than 850μm, a positive pulse signal will be generated.” Why is a positive pulse signal generated when particle size is larger than 850μm? There still need more theories for the results.

13. The English writing should be improved.
